# Learning Barrier Certificates: Towards Safe Reinforcement Learning with Zero Training-time Violations

**Yuping Luo**
Princeton University
yupingl@cs.princeton.edu

**Tengyu Ma**
Stanford University
tengyuma@stanford.edu

## Abstract

Training-time safety violations have been a major concern when we deploy reinforcement learning algorithms in the real world. This paper explores the possibility of safe RL algorithms with zero training-time safety violations in the challenging setting where we are only given a safe but trivial-reward initial policy without any prior knowledge of the dynamics and additional offline data. We propose an algorithm, **Co-tra**ined **B**arrier Certificate for **S**afe RL (CRABS), which iteratively *learns* barrier certificates, dynamics models, and policies. The barrier certificates are learned via adversarial training and ensure the policy's safety assuming calibrated learned dynamics. We also add a regularization term to encourage larger certified regions to enable better exploration. Empirical simulations show that zero safety violations are already challenging for a suite of simple environments with only 2-4 dimensional state space, especially if high-reward policies have to visit regions near the safety boundary. Prior methods require hundreds of violations to achieve decent rewards on these tasks, whereas our proposed algorithms incur zero violations.

## 1 Introduction

Researchers have demonstrated that reinforcement learning (RL) can solve complex tasks such as Atari games [Mnih et al., 2015], Go [Silver et al., 2017], dexterous manipulation tasks [Akkaya et al., 2019], and many more robotics tasks in simulated environments [Haarnoja et al., 2018]. However, deploying RL algorithms to real-world problems still faces the hurdle that they require many unsafe environment interactions. For example, a robot's unsafe environment interactions include falling and hitting other objects, which incur physical damage costly to repair. Many recent deep RL works reduce the number of environment interactions significantly (e.g., see Haarnoja et al. [2018], Fujimoto et al. [2018], Janner et al. [2019], Dong et al. [2020], Luo et al. [2019], Chua et al. [2018] and reference therein), but the number of unsafe interactions is still prohibitive for safety-critical applications such as robotics, medicine, or autonomous vehicles [Berkenkamp et al., 2017].

Reducing the number of safety violations may not be sufficient for these safety-critical applications—we may have to eliminate them. This paper explores the possibility of safe RL algorithms with *zero safety violations* in both training time and test time. We also consider the challenging setting where we are only given a safe but trivial-reward initial policy.

A recent line of works on safe RL design novel actor-critic based algorithms under the constrained policy optimization formulation [Thananjeyan et al., 2021, Srinivasan et al., 2020, Bharadhwaj et al., 2020, Yang et al., 2020, Stooke et al., 2020]. They significantly reduce the number of training-time safety violations. However, these algorithms fundamentally learn the safety constraints by contrasting the safe and unsafe trajectories. In other words, because the safety set is only specified through the

35th Conference on Neural Information Processing Systems (NeurIPS 2021).

safety costs that are observed *postmortem*, the algorithms only learn the concept of safety through seeing unsafe trajectories. Therefore, these algorithms cannot achieve zero training-time violations. For example, even for the simple 2D inverted pendulum environment, these methods still require at least 80 unsafe trajectories (see Figure 2 in Section 6).

Another line of work utilizes ideas from control theory and model-based approach [Cheng et al., 2019, Berkenkamp et al., 2017, Taylor et al., 2019, Zeng et al., 2020]. These works propose sufficient conditions involving certain Lyapunov functions or control barrier functions that can certify the safety of a subset of states or policies [Cheng et al., 2019]. These conditions assume access to calibrated dynamical models. They can, in principle, permit safety guarantees without visiting any unsafe states because, with the calibrated dynamics, we can foresee future danger. However, control barrier functions are often non-trivially *handcrafted* with prior knowledge of the environments [Ames et al., 2019, Nguyen and Sreenath, 2016].

This work aims to design model-based safe RL algorithms that achieve zero training-time safety violations by *learning* the barrier certificates *iteratively*. We present the algorithm **Co-tra**ined **B**arrier Certificate for **S**afe RL (CRABS), which alternates between *learning* barrier certificates that certify the safety of *larger* regions of states, optimizing the policy, collecting more data within the certified states, and refining the learned dynamics with data.

The work of Richards et al. [2018] is a closely related prior result, which learns a Lyapunov function given a fixed dynamics model via discretization of the state space. Our work significantly extends it with three algorithmic innovations. First, we use adversarial training to learn the certificates, which avoids discretizing state space and can potentially work with higher dimensional state space than the two-dimensional problems in Richards et al. [2018]. Second, we do not assume a given, globally accurate dynamics; instead, we learn the dynamics from safe explorations. We achieve this by co-learning the certificates, dynamics, and policy to iteratively grow the certified region and improve the dynamics and still maintain zero violations. Thirdly, the work Richards et al. [2018] only certifies the safety of some states and does not involve learning a policy. In contrast, our work learns a policy and tailors the certificates to the learned policies. In particular, our certificates aim to certify only states near the trajectories of the current and past policies—this allows us to not waste the expressive power of the certificate parameterization on irrelevant low-reward states.

We evaluate our algorithms on a suite of tasks, including a few where achieving high rewards requires careful exploration near the safety boundary. For example, in the ***Swing*** environment, the goal is to swing a rod with the largest possible angle under the safety constraints that the angle is less than $90°$. We show that our method reduces the number of safety violations from several hundred to zero on these tasks.

## 2 Setup and Preliminaries

### 2.1 Problem Setup

We consider the standard RL setup with an infinite-horizon *deterministic* Markov decision process (MDP). An MDP is specified by a tuple $(\mathcal{S}, \mathcal{A}, \gamma, r, \mu, T)$, where $\mathcal{S}$ is the state space, $\mathcal{A}$ is the action space, $r : \mathcal{S} \times \mathcal{A} \to \mathbb{R}$ is the reward function, $0 \leq \gamma < 1$ is the discount factor, $\mu$ is the distribution of the initial state, and $T : \mathcal{S} \times \mathcal{A} \to \mathcal{S}$ is the deterministic dynamics model. Let $\Delta(\mathcal{X})$ denote the family of distributions over a set $\mathcal{X}$. The expected discounted total reward of a policy $\pi : \mathcal{S} \to \Delta(\mathcal{A})$ is defined as

$$J(\pi) = \mathbb{E} \left[ \sum_{i=0}^{\infty} \gamma^i r(s_i, a_i) \right],$$

where $s_0 \sim \mu, a_i \sim \pi(s_i), s_{i+1} = T(s_i, a_i)$ for $i \geq 0$. The goal is to find a policy $\pi$ which maximizes $J(\pi)$.

Let $\mathcal{S}_{\text{unsafe}} \subset \mathcal{S}$ be the set of unsafe states specified by the user. The user-specified safe set $\mathcal{S}_{\text{safe}}$ is defined as $\mathcal{S} \backslash \mathcal{S}_{\text{unsafe}}$. A state $s$ is (user-specified) safe if $s \in \mathcal{S}_{\text{safe}}$. A trajectory is safe if and only if all the states in the trajectory are safe. An initial state drawn from $\mu$ is assumed to safe with probability 1. We say a deterministic policy $\pi$ is safe starting from state $s$, if the infinite-horizon trajectory obtained by executing $\pi$ starting from $s$ is safe. We also say a policy $\pi$ is safe if it is safe starting from an initial state drawn from $\mu$ with probability 1. A major challenge toward safe RL is the existence of

irrecoverable states which are currently safe but will eventually lead to unsafe states regardless of future actions. We define the notion formally as follows.

**Definition 1.** *A state $s$ is* viable *iff there exists a policy $\pi$ such that $\pi$ is safe starting from $s$, that is, executing $\pi$ starting from $s$ for infinite steps never leads to an unsafe state. A user-specified safe state that is not viable is called an* irrecoverable *state.*

We remark that unlike Srinivasan et al. [2020], Roderick et al. [2020], we do not assume all safe states are viable. We rely on the extrapolation and calibration of the dynamics to foresee risks. A calibrated dynamics model $\widehat{\mathcal{T}}$ predicts a confidence region of states $\widehat{\mathcal{T}}(s, a) \subseteq \mathcal{S}$, such that for any state $s$ and action $a$, we have $T(s, a) \in \widehat{\mathcal{T}}(s, a)$.

## 2.2 Preliminaries on Barrier Certificate

Barrier certificates are powerful tools to certify the stability of a dynamical system. Barrier certificates are often applied to a continuous-time dynamical system, but here we describe its discrete-time version where our work is based upon. We refer the readers to Prajna and Jadbabaie [2004], Prajna and Rantzer [2005] for more information about continuous-time barrier certificates.

Given a discrete-time dynamical system $s_{t+1} = f(s_t)$ *without control* starting from $s_0$, a function $h : \mathcal{S} \to \mathbb{R}$ is a barrier certifcate if for any $s \in \mathcal{S}$ such that $h(s) \geq 0$, $h(f(s)) \geq 0$. Zeng et al. [2020] considers a more restrictive requirement: For any state $s \in \mathcal{S}$, $h(f(s)) \geq \alpha h(s)$ for a constant $0 \leq \alpha < 1$.

it is easy to use a barrier certificate $h$ to show the stability of the dynamical system. Let $\mathcal{C}_h = \{s : h(s) \geq 0\}$ be the superlevel set of $h$. The requirement of barrier certificates directly translates to the requirement that if $s \in \mathcal{C}_h$, then $f(s) \in \mathcal{C}_h$. This property of $\mathcal{C}_h$, which is known as the *forward-invariant* property, is especially useful in safety-critical settings: suppose a barrier certificate $h$ such that $\mathcal{C}_h$ does not contain unsafe states and contains the initial state $s_0$, then it is guaranteed that $\mathcal{C}_h$ contains the entire trajectory of states $\{s_t\}_{t \geq 0}$ which are safe.

Finding barrier certificates requires a known dynamics $f$, which often can only be approximated in practice. This issue can be resolved by using a well-calibrated dynamics model $\hat{f}$, which predicts a confidence interval containing the true output. When a calibrated dynamics model $\hat{f}$ is used, we require that for any $s \in \mathcal{S}$, $\min_{s' \in \hat{f}(s)} h(s') \geq 0$.

Control barrier functions [Ames et al., 2019] are extensions to barrier certificates in the control setting. That is, control barrier functions are often used to *find* an action to meet the safety requirement instead of certifying the stability of a closed dynamical system. In this work, we simply use barrier certificates because in Section 3, we view the policy and the calibrated dynamics model as a whole closed dynamical system whose stability we are going to certify.

## 3 Learning Barrier Certificates via Adversarial Training

This section describes an algorithm that learns a barrier certificate for a fixed policy $\pi$ under a calibrated dynamics model $\widehat{\mathcal{T}}$. Concretely, to certify a policy $\pi$ is safe, we aim to learn a (discrete-time) barrier certificate $h$ that satisfies the following three requirements.

**R.1.** For $s_0 \sim \mu$, $h(s_0) \geq 0$ with probability 1.
**R.2.** For every $s \in \mathcal{S}_{\text{unsafe}}$, $h(s) < 0$.
**R.3.** For any $s$ such that $h(s) \geq 0$, $\min_{s' \in \widehat{\mathcal{T}}(s, \pi(s))} h(s) \geq 0$.

Requirement **R.1** and **R.3** guarantee that the policy $\pi$ will never leave the set $\mathcal{C}_h = \{s \in \mathcal{S} : h(s) \geq 0\}$ by simple induction. Moreover, **R.2** guarantees that $\mathcal{C}_h$ only contains safe states and therefore the policy never visits unsafe states.

In the rest of the section, we aim to design and train such a barrier certificate $h = h_\phi$ parametrized by neural network $\phi$.

$h_\phi$ **parametrization.**   The three requirements for a barrier certificate are challenging to simultaneously enforce with constrained optimization involving neural network parameterization. Instead, we will parametrize $h_\phi$ with **R.1** and **R.2** built-in such that for any $\phi$, $h_\phi$ always satisfies **R.1** and **R.2**.

We assume the initial state $s_0$ is deterministic (the parameterization can be extended to multiple initial states.) To capture the known user-specified safety set, we first handcraft a continuous function $\mathcal{B}_{\text{safe}} : \mathcal{S} \to \mathbb{R}_{\geq 0}$ satisfying $\mathcal{B}_{\text{safe}}(s) \approx 0$ for typical $s \in \mathcal{S}_{\text{safe}}$ and $\mathcal{B}_{\text{safe}}(s) > 1$ for any $s \in \mathcal{S}_{\text{unsafe}}$.[1] The construction of $\mathcal{B}_{\text{safe}}$ does not need prior knowledge of irrecoverable states, but only the user-specified safety set $\mathcal{S}_{\text{safe}}$. To further encode the user-specified safety set into $h_\phi$, we choose $h_\phi$ to be of form $h_\phi(s) = 1 - \text{Softplus}(f_\phi(s) - f_\phi(s_0)) - \mathcal{B}_{\text{safe}}(s)$, where $f_\phi$ is a neural network, and $\text{Softplus}(x) = \log(1 + e^x)$.

Because $s_0$ is safe and $\mathcal{B}_{\text{safe}}(s_0) \approx 0$, $h_\phi(s_0) \approx 1 - \text{Softplus}(0) > 0$. Therefore $h_h$ satisfies **R.1**. Moreover, for any $s \in \mathcal{S}_{\text{unsafe}}$, we have $h_\phi(s) < 1 - \mathcal{B}_{\text{safe}}(s) < 0$, so $h_\phi$ in our parametrization satisfies **R.2** by design.

**Training barrier certificates.** We now move on to training $\phi$ to satisfy **R.3**. Let

$$U(s, a, h) := \max_{s' \in \widehat{\mathcal{T}}(s,a)} -h(s'). \tag{1}$$

Then, **R.3** requires $U(s, \pi(s), h_\phi) \leq 0$ for *any* $s \in \mathcal{C}_{h_\phi}$, The constraint in **R.3** naturally leads up to formulate the problem as a min-max problem. Define our objective function to be

$$C^*(h_\phi, U, \pi) := \max_{s \in \mathcal{C}_{h_\phi}} U(s, \pi(s), h_\phi) = \max_{s \in \mathcal{C}_{h_\phi}, s' \in \widehat{\mathcal{T}}(s,\pi(s))} -h(s'), \tag{2}$$

and we want to minimize $C^*$ w.r.t. $\phi$:

$$\min_\phi C^*(h_\phi, U, \pi) = \min_\phi \max_{s \in \mathcal{C}_{h_\phi}, s' \in \widehat{\mathcal{T}}(s,\pi(s))} -h(s'), \tag{3}$$

Our goal is to ensure the minimum value is less than 0. We use gradient descent to solve the optimization problem. We also derive the gradient of $C^*(L_\phi, U, \pi)$ w.r.t. $\phi$ :

$$\nabla_\phi C^*(h_\phi, U, \pi) = \nabla_\phi U(s^*, \pi(s^*), h_\phi) + \frac{\|\nabla_\phi U(s^*, \pi(s^*), h_\phi)\|_2}{\|\nabla_\phi h_\phi(s^*)\|_2} \nabla_\phi h_\phi(s^*), \tag{4}$$

where $s^* := \arg\max_{s:h_\phi(s) \leq 1} U(s, \pi(s), h_\phi)$ and we defer the derivation to Appendix A.

**Computing the adversarial $s^*$.**

Equation (4) requires us to compute $s^*$ efficiently. Because the maximization problem with respect to $s$ is nonconcave, there could be multiple local maxima. In practice, we find that it is more efficient and reliable to use multiple local maxima to compute $\nabla_\phi C^*$ and then average the gradient.

Solving $s^*$ is highly non-trivial, as it is a non-concave optimization problem with a constraint $s \in \mathcal{C}_{h_\phi}$. To deal with the constraint, we introduce a Lagrangian multiplier $\lambda$ and optimize $U(s, \pi(s), h_\phi) - \lambda \mathbb{I}_{s \in \mathcal{C}_{h_\phi}}$ w.r.t. $s$ without any constraints. However, it is still very time-consuming to solve an optimization problem independently at each time. Based on the observation that the parameters of $h$ do not change too much by one step of gradient step, we can use the optimal solution from the last optimization problem as the initial solution for the next one, which naturally leads to the idea of maintaining a set of candidates of $s^*$'s during the computation of $\nabla_\phi C^*$.

We use Metropolis-adjusted Langevin algorithm (MALA) to maintain a set of candidates $\{s_1, \ldots, s_m\}$ which are supposed to sample from $\exp(\tau(U(s, \pi(s), h_\phi) - \lambda \mathbb{I}_{s \in \mathcal{C}_{h_\phi}}))$. Here $\tau$ is the temperature indicating we want to focus on the samples with large $U(s, \pi(s), h_\phi)$. Although the indicator function always have zero gradient, it is still useful in the sense that MALA will reject $s_i \notin \mathcal{C}_{h_\phi}$. A detailed description of MALA is given in Appendix D.

We choose MALA over gradient descent because the maintained candidates are more diverse, approximate local maxima. If we use gradient descent to find $s^*$, then multiple runs of GD likely arrive at the same $s^*$, so that we lost the parallelism from simultaneously working with multiple

---

[1]The function $\mathcal{B}_{\text{safe}}(s)$ is called a barrier function for the user-specified safe set in the optimization literature. Here we do not use this term to avoid confusion with the barrier certificate.

---

**Algorithm 1** Learning barrier certificate $h_\phi$ for a policy $\pi$ w.r.t. a calibrated dynamics model $\widehat{\mathcal{T}}$.

---

**Require:** Temperature $\tau$, Lagrangian multiplier $\lambda$, and optionally a regularization function Reg.
1: Let $U$ be defined as in Equation (1).
2: Initialize $m$ candidates of $s_1, \ldots, s_m \in \mathcal{S}$ randomly.
3: **for** $n$ iterations **do**
4:     **for** every candidate $s_i$ **do**
5:         sample $s_i \sim \exp(\tau U(s, \pi(s), h_\phi) - \lambda \mathbb{I}_{s \in \mathcal{C}_h})$ by MALA (Algorithm 5).
6:     $W \leftarrow \{s_i : h_\phi(s_i) \geq 0, i \in [m]\}$.
7:     Train $\phi$ to minimize $C^*(h_\phi, U, \pi) + \text{Reg}(\phi)$ using all candidates in $W$.

---

**Algorithm 2** CRABS: **C**o-t**ra**ined **B**arrier Certificate for **S**afe RL (Details in Section 4)

---

**Require:** An initial safe policy $\pi_{\text{init}}$.
1: Collected trajectories buffer $\widehat{D} \leftarrow \emptyset$; $\pi \leftarrow \pi_{\text{init}}$.
2: **for** $T$ epochs **do**
3:     Invoke Algorithm 3 to safely collect trajectories (using $\pi$ as the safeguard policy and a noisy version of $\pi$ as the $\pi^{\text{expl}}$). Add the trajectories to $\widehat{D}$.
4:     Learn a calibrated dynamics $\widehat{\mathcal{T}}$ with $\widehat{D}$.
5:     Learn a barrier certificate $h$ that certifies $\pi$ w.r.t. $\widehat{\mathcal{T}}$ using Algorithm 1 with regularization.
6:     Optimize policy $\pi$ (according to the reward), using data in $\widehat{D}$, with the constraint that $\pi$ is certified by $h$.

---

local maxima. MALA avoids this issue by its intrinsic stochasticity, which can also be controlled by adjusting the hyperparameter $\tau$.

We summarize our algorithm of training barrier certificates in Algorithm 1 (which contains optional regularization that will be discussed in Section 4.2). At Line 2, the initialization of $s_i$'s is arbitrary, as long as they have a sort of stochasticity.

## 4 CRABS: Co-trained Barrier Certificate for Safe RL

In this section, we present our main algorithm, **C**o-t**ra**ined **B**arrier Certificate for **S**afe RL (CRABS), shown in Algorithm 2, to *iteratively* co-train barrier certificates, policy and dynamics, using the algorithm in Section 3. In addition to parametrizing $h$ by $\phi$, we further parametrize the policy $\pi$ by $\theta$, and parametrize calibrated dynamics model $\widehat{\mathcal{T}}$ by $\omega$. CRABS alternates between training a barrier certificate that certifies the policy $\pi_\theta$ w.r.t. a calibrated dynamics model $\widehat{\mathcal{T}}_\omega$ (Line 5), collecting data safely using the certified policy (Line 3, details in Section 4.1), learning a calibrated dynamics model (Line 4, details in Section 4.3), and training a policy with the constraint of staying in the superlevel set of the barrier function (Line 6, details in Section 4.4). In the following subsections, we discuss how we implement each line in detail.

### 4.1 Safe Exploration with Certified Safeguard Policy

Safe exploration is challenging because it is difficult to detect irrecoverable states. The barrier certificate is designed to address this — a policy $\pi$ certified by some $h$ guarantees to stay within $\mathcal{C}_h$ and therefore can be used for collecting data. However, we may need more diversity in the collected data beyond what can be offered by the deterministic certified policy $\pi^{\text{safeguard}}$. Thanks to the contraction property **R.3**, we in fact know that any exploration policy $\pi^{\text{expl}}$ within the superlevel set $\mathcal{C}_h$ can be made safe with $\pi^{\text{safeguard}}$ being a safeguard policy—we can first try actions from $\pi^{\text{expl}}$ and see if they stay within the viable subset $\mathcal{C}_h$, and if none does, invoke the safeguard policy $\pi^{\text{safeguard}}$. Algorithm 3 describes formally this simple procedure that makes any exploration policy $\pi^{\text{expl}}$ safe. By a simple induction, one can see that the policy defined in Algorithm 3 maintains that all the visited states lie in $\mathcal{C}_h$.

The safeguard policy $\pi^{\text{safeguard}}$ is supposed to safeguard the exploration. However, activating the safeguard too often is undesirable, as it only collects data from $\pi^{\text{safeguard}}$ so there will be little

**Algorithm 3** Safe exploration with safeguard policy $\pi^{\text{safeguard}}$

---

**Require:** (1) A policy $\pi^{\text{safeguard}}$ certified by barrier certificate $h$, (2) any proposal exploration policy $\pi^{\text{expl}}$.
**Require:** A state $s \in \mathcal{C}_{h_\phi}$.
1: Sample $n$ actions $a_1, \ldots a_n$ from $\pi^{\text{expl}}(s)$.
2: **if** there exists an $a_i$ such that $U(s, a_i, h) \leq 1$ **then**
3:     **return:** $a_i$
4: **else**
5:     **return:** $\pi^{\text{safeguard}}(s)$.

---

exploration. To mitigate this issue, we often choose $\pi^{\text{expl}}$ to be a noisy version of $\pi^{\text{safeguard}}$ so that $\pi^{\text{expl}}$ will be roughly safe by itself. Moreover, the safeguard policy $\pi^{\text{safeguard}}$ will be trained via optimizing the reward function as shown in the next subsections. Therefore, a noisy version of $\pi^{\text{safeguard}}$ will explore the high-reward region and avoid unnecessary exploration.

Following Haarnoja et al. [2018], the policy $\pi_\theta$ is parametrized as $\tanh(\mu_\theta(s))$, and the proposal exploration policy $\pi_\theta^{\text{expl}}$ is parametrized as $\tanh(\mu_\theta(s) + \sigma_\theta(s)\zeta)$ for $\zeta \sim \mathcal{N}(0, I)$, where $\mu_\theta$ and $\sigma_\theta$ are two neural networks. Here the $\tanh$ is applied to squash the outputs to the action set $[-1, 1]$.

## 4.2 Regularizing Barrrier Certificates

The quality of exploration is directly related to the quality of policy optimization. In our case, the exploration is only within the learned viable set $\mathcal{C}_{h_\phi}$ and it will be hindered if $\mathcal{C}_{h_\phi}$ is too small or does not grow during training. To ensure a large and growing viable subset $\mathcal{C}_{h_\phi}$, we encourage the volume of $\mathcal{C}_{h_\phi}$ to be large by adding a regularization term

$$\text{Reg}(\phi; \hat{h}) = \mathbb{E}_{s \in \mathcal{S}}[\text{relu}(\hat{h}(s) - h_\phi(s))],$$

Here $\hat{h}$ is the barrier certificate obtained in the previous epoch. In the ideal case when $\text{Reg}(\phi; \hat{h}) = 0$, we have $\mathcal{C}_{h_\phi} \supset \mathcal{C}_{\hat{h}}$, that is, the new viable subset $\mathcal{C}_{h_\phi}$ is at least bigger than the reference set (which is the viable subset in the previous epoch.) We compute the expectation over $\mathcal{S}$ approximately by using the set of candidate $s$'s maintained by MALA.

In summary, to learn $h_\phi$ in CRABS, we minimize the following objective (for a small positive constant $\lambda$) over $\phi$ as shown in Algorithm 1:

$$\mathcal{L}(\phi; U, \pi_\theta, \hat{h}) = C^*(L_\phi, U, \pi_\theta) + \lambda \text{Reg}(\phi; \hat{h}). \tag{5}$$

We remark that the regularization is not the only reason why the viable set $\mathcal{C}_{h_\phi}$ can grow. When the dynamics becomes more accurate as we collect more data, the $\mathcal{C}_{h_\phi}$ will also grow. This is because an inaccurate dynamics will typically make the $\mathcal{C}_{h_\phi}$ smaller—it is harder to satisfy **R.3** when the confidence region $\widehat{\mathcal{T}}(s, \pi(s))$ in the constraint contains many possible states. Vice versa, shrinking the size of the confidence region will make it easier to certify more states.

## 4.3 Learning a Calibrated Dynamics Model

It is a challenging open question to obtain a dynamics model $\widehat{\mathcal{T}}$ (or any supervised learning model) that is theoretically well-calibrated especially with domain shift [Zhao et al., 2020]. In practice, we heuristically approximate a calibrated dynamics model by learning an ensemble of probabilistic dynamics models, following common practice in RL [Yu et al., 2020, Janner et al., 2019, Chua et al., 2018]. We learn $K$ probabilistic dynamics models $f_{\omega_1}, \ldots, f_{\omega_K}$ using the data in the replay buffer $\widehat{D}$. (Interestingly, prior work shows that an ensemble of probabilistic models can still capture the error of estimating a deterministic ground-truth dynamics [Janner et al., 2019, Chua et al., 2018].) Each probabilistic dynamics model $f_{\omega_i}$ outputs a Gaussian distribution $\mathcal{N}(\mu_{\omega_i}(s, a), \text{diag}(\sigma_{\omega_i}^2(s, a)))$ with diagonal covariances, where $\mu_{\omega_i}$ and $\sigma_{\omega_i}$ are parameterized by neural networks. Given a replay buffer $\widehat{D}$, the objective for a probabilistic dynamics model $f_{\omega_i}$ is to minimize the negative log-likelihood:

$$\mathcal{L}_{\widehat{\mathcal{T}}}(\omega_i) = -\mathbb{E}_{(s, a, s') \sim \widehat{D}} \left[ -\log f_{\omega_i}(s'|s, a) \right]. \tag{6}$$

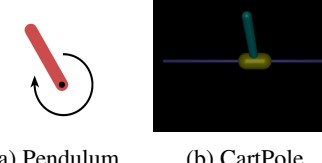

(a) Pendulum      (b) CartPole

Figure 1: Illustration of environments. The left figure illustrates the Pendulum environment, which is used by *Upright* and *Tilt* tasks. The right figer illustrates the CartPole environment, which is used by *Move* and *Swing* tasks.

The only difference in the training procedure of these probabilistic models is the randomness in the initialization and mini-batches. We simply aggregate the means of all learn dynamics models as a coarse approximation of the confidence region, i.e., $\widehat{\mathcal{T}}(s,a) = \{\mu_{\omega_i}(s,a)\}_{i \in [K]}$.

### 4.4 Policy Optimization

We describe our policy optimization algorithm in Algorithm 4 . The desiderata here are (1) the policy needs certified by the current barrier certificate $h$ and (2) the policy has as high reward as possible. We break down our policy optimization algorithm into two components: First, we optimize the total rewards $J(\pi_\theta)$ of the policy $\pi_\theta$; Second, we use adversarial training to guarantee the optimized policy can be certified by $h_\phi$. The modification of SAC is to some extent non-essential and mostly for technical convenience of making SAC somewhat compatible with the constraint set. Instead, it is the adversarial step that fundamentally guarantees that the policy is certified by the current $h_\phi$.

**Adversarial training.** We use adversarial training to guarantee $\pi_\theta$ can be certified by $h_\phi$. Similar to what we've done in training $h_\phi$ adversarially, the objective for training $\pi_\theta$ is to minimize $C^*(h_\phi, U, \pi_\theta)$. Unlike the case of $\phi$, the gradient of $C^*(h_\phi, U, \pi_\theta)$ w.r.t. $\theta$ is simply $\nabla_\theta U(s^*, \pi_\theta(s^*), h_\phi)$, as the constraint $h_\phi(s)$ is unrelated to $\pi_\theta$. We also use MALA to solve $s^*$ and plug it into the gradient term $\nabla_\theta U(s^*, \pi_\theta(s^*), h_\phi)$.

**Optimizing $J(\pi_\theta)$.** We use a modified SAC [Haarnoja et al., 2018] to optimize $J(\pi_\theta)$. As the modification is for safety concerns and is minor, we defer it to Appendix B. As a side note, although we only optimize $\pi_\theta^{\text{expl}}$ here, $\pi_\theta$ is also optimized implicitly because $\pi_\theta^{\text{expl}}$ simply outputs the mean of $\pi_\theta$ deterministically.

## 5 High-risk, High-reward Environments

We design four tasks, three of which are high-risk, high-reward tasks, to check the efficacy of our algorithm. Even though they are all based on inverted pendulum or cart pole, we choose the reward function to be somewhat conflicted with the safety constraints. That is, the optimal policy needs to take a trajectory that is near the safety boundary. This makes the tasks particularly challenging and suitable for stress testing our algorithm's capability of avoiding irrecoverable states.

These tasks have state dimension dimensions between 2 to 4. We focus on the relatively low dimensional environments to avoid conflating the failure to learn accurate dynamics models from data and the failure to provide safety given a learned approximate dynamics. Indeed, we identify that the major difficulty to scale up to high-dimensional environments is that it requires significantly more data to learn a decent high-dimensional dynamics that can predict long-horizon trajectories. We remark that we aim to have zero violations. This is very difficult to achieve, even if the environment is low dimensional. As shown by Section 6, many existing algorithms fail to do so.

(a) *Upright*. The task is based on Pendulum-v0 in Open AI Gym [Brockman et al., 2016], as shown in Figure 1a. The agent can apply torque to control a pole. The environment involves the crucial quantity: the tilt angle $\theta$ which is defined to be the angle between the pole and a vertical line. The safety requirement is that the pole does not fall below the horizontal line. Technically, the user-specified safety set is $\{\theta : |\theta| \leq \theta_{\max} = 1.5\}$ (note that the threshold is very close to $\frac{\pi}{2}$ which corresponds to 90°.) The reward function $r$ is $r(s,a) = -\theta^2$, so the optimal policy minimizes the angle and angular speed by keeping the pole upright. The horizon is 200 and the initial state $s_0 = (0.3, -0.9)$.

(b) *Tilt*. This action set, dynamics, and horizon, and safety set are the same as in *Upright*. The reward function is different: $r(s,a) = -(\theta_{\text{limit}} - \theta)^2$. The optimal policy is supposed to stay tilting near the

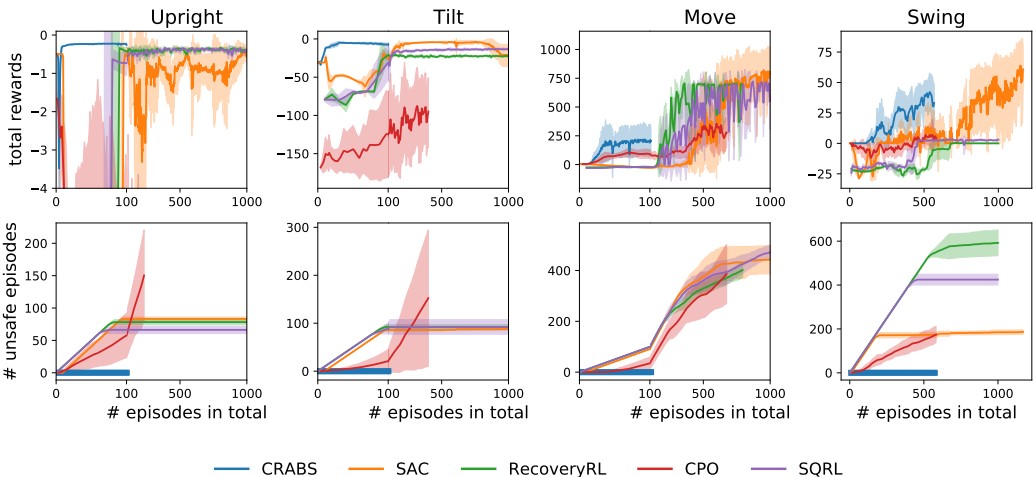

Figure 2: Comparision between CRABS and baselines. CRABS can learn a policy without any safety violations, while other baselines have a lot of safety violations. We run each algorithm four times with independent randomness. The solid curves indicate the mean of four runs and the shaded areas indicate one standard deviation around the mean.

angle $\theta = \theta_{\text{limit}}$ where $\theta_{\text{limit}} = -0.41151684$ is the largest angle the pendulum can stay balanced. The challenge is during exploration, it is easy for the pole to overshoot and violate the safety constraints.

(c) *Move*. The task is based on a cart pole and the goal is to move a cart (the yellow block) to control the pole (with color teal), as shown in Figure 1b. The cart has an $x$ position between $-1$ and $1$, and the pole also has an angle $\theta \in [-\frac{\pi}{2}, \frac{\pi}{2}]$ with the same meaning as *Upright* and *Tilt*. The starting position is $x = \theta = 0$. We design the reward function to be $r(s, a) = x^2$. The user-specified safety set is $\{(x, \theta) : |\theta| \leq \theta_{\max} = 0.2, |x| \leq 0.9\}$ where 0.2 corresponds to roughly $11°$. Therefore, the optimal policy needs to move the cart and the pole slowly in one direction, preventing the pole from falling down and the cart from going too far. The horizon is set to 1000.

(d) *Swing*. This task is similar to *Move*, except for a few differences: The reward function is $r(s, a) = \theta^2$; The user-specified safety set is $\{(x, \theta) : |\theta| \leq \theta_{\max} = 1.5, |x| \leq 0.9\}$. So the optimal policy will swing back and forth to some degree and needs to control the angles well so that it does not violate the safety requirement.

For all the tasks, once the safety constraint is violated, the episode will terminate immediately and the agent will receive a reward of -30 as a penalty. The number -30 is tuned by running SAC and choosing the one that SAC performs best with.

## 6 Experimental Results

In this section, we conduct experiments to answer the following question: Can CRABS learn a reasonable policy without safety violations in the designed tasks?

**Baselines.** We compare our algorithm CRABS against four baselines: (a) **Soft Actor-Critic** (SAC) [Haarnoja et al., 2018], one of the state-of-the-art RL algorithms, (b) **Constrained Policy Optimization** (CPO) [Achiam et al., 2017], a safe RL algorithm which builds a trust-region around the current policy and optimizes the policy in the trust-region, (c) **RecoveryRL** [Thananjeyan et al., 2021] which leverages offline data to pretrain a risk-sensitive $Q$ function and also utilize two policies to achieving two goals (being safe and obtaining high rewards), and (d) **SQRL** [Srinivasan et al., 2020] which leverages offline data in an easier environment and fine-tunes the policy in a more difficult environment. SAC and CPO are given an initial safe policy for safe exploration, while RecoveryRL and SQRL are given offline data containing 40K steps from both mixed safe and unsafe trajectories which are free and are not counted. CRABS collects more data at each iteration in *Swing* than in other tasks to learn a better dynamics model $\widehat{\mathcal{T}}$. For SAC, we use the default hyperparameters because

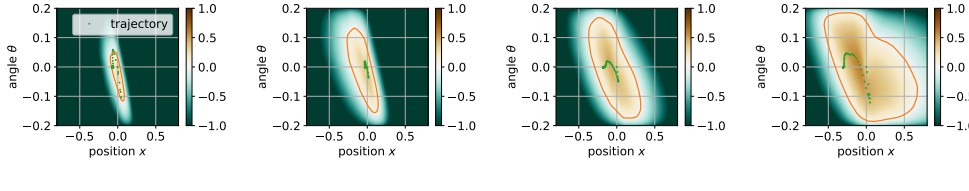

(a) $\mathcal{C}_{h_\phi}$ after 0 epochs    (b) $\mathcal{C}_{h_\phi}$ after 5 epochs    (c) $\mathcal{C}_{h_\phi}$ after 10 epochs    (d) $\mathcal{C}_{h_\phi}$ after 15 epochs

Figure 3: Visualization of the growing viable subsets learned by CRABS in *Move*. To illustrate the 4-dimensional state space, we project a state from $[x, \theta, \dot{x}, \dot{\theta}]$ to $[x, \theta]$. The red curve encloses superlevel set $\mathcal{C}_{h_\phi}$, while the green points indicate the projected trajectory of the current safe policy. We can also observe that policy $\pi$ learns to move left as required by the task. We note that shown states in the trajectory sometimes seemingly are not be enclosed by the red curve due to the projection.

we found they are not sensitive. For RecoveryRL and SQRL, the hyperparameters are tuned in the same way as in Thananjeyan et al. [2021] . For CPO, we tune the step size and batch size. More details of experiment setup and the implementation of baselines can be found in Appendix C.

**Results.** Our main results are shown in Figure 2. From the perspective of total rewards, SAC achieves the best total rewards among all of the 5 algorithms in *Move* and *Swing*. In all tasks, CRABS can achieve reasonable total rewards and learns faster at the beginning of training, and we hypothesize that this is directly due to its strong safety enforcement. RecoveryRL and SQRL learn faster than SAC in *Move*, but they suffer in *Swing*. RecoveryRL and SQRL are not capable of learning in *Swing*, although we observed the average return during exploration at the late stages of training can be as high as 15. CPO is quite sample-inefficient and does not achieve reasonable total rewards as well.

From the perspective of safety violations, CRABS surpasses all baselines **without a single safety violation**. The baseline algorithms always suffer from many safety violations. SAC, SQRL, and RecoveryRL have a similar number of unsafe trajectories in *Upright, Tilt, Move*, while in *Swing*, SAC has the fewest violations and RecoveryRL has the most violations. CPO has a lot of safety violations. We observe that for some random seeds, CPO does find a safe policy and once the policy is trained well, the safety violations become much less frequent, but for other random seeds, CPO keeps visiting unsafe trajectories before it reaches its computation budget.

**Visualization of learned viable subset $\mathcal{C}_{h_\phi}$.** We visualized the viable set $\mathcal{C}_{h_\phi}$ in Figure 3. As shown in the figure, our algorithm CRABS succeeds in certifying more and more viable states and does not get stuck locally, which demonstrates the efficacy of the regularization at Section 4.2.

**Handcrafted barrier function.** To demonstrate the advantage of learning a barrier function, we also conduct experiments on a variant of CRABS, which uses a handcrafted barrier certificate by ourselves and does not train it, that is, Algorithm 2 without Line 5. Our goal for the handcrafted barrier function is to only show that it's non-trivial to find a subset of viable states by handcrafting a region and therefore to demonstrate the necessity to learn the barrier certificates. In the experiments, we handcrafted a function $g$, whose superlevel set is a subset of safe states, and restricted exploration in this region (the superlevel set of $g$) in a similar manner as our current exploration scheme (Algorithm 3). Here only satisfies **R.1** and **R.2** but not necessarily **R.3**. Therefore, strictly speaking, the designed function $g$ is not a barrier certificate, but only merely attempts to characterize a subset of safe states and restrict the exploration to it. More details of the handcrafted function $g$ are available at Appendix C.

The results show that this variant does not perform well: It does not achieve high rewards, and has many safety violations. We hypothesize that the policy optimization is often burdened by adversarial training, and the safeguard policy sometimes cannot find an action to stay within the superlevel set $\mathcal{C}_h$.

# 7 Related Work

Prior works about Safe RL take very different approaches. Dalal et al. [2018] adds an additional layer, which corrects the output of the policy locally. Some of them use Lagrangian methods to solve CMDP, while the Lagrangian multiplier is controlled adaptively [Tessler et al., 2018] or by a PID [Stooke et al., 2020]. Achiam et al. [2017], Yang et al. [2020] build a trust-region around the current policy. Eysenbach et al. [2017] learns a reset policy so that the policy only explores the states that can go back to the initial state. Turchetta et al. [2020] introduces a learnable teacher, which keeps the student safe and helps the student learn faster in a curriculum manner. Srinivasan et al. [2020] pre-trains a policy in a simpler environment and fine-tunes it in a more difficult environment. Bharadhwaj et al. [2020] learns conservative safety critics which underestimate how safe the policy is, and uses the conservative safety critics for safe exploration and policy optimization. Thananjeyan et al. [2021] makes use of existing offline data and co-trains a recovery policy.

Another line of work involves Lyapunov functions and barrier functions. Donti et al. [2020] constructs sets of stabilizing actions using a Lyapunov function, and project the action to the set, while Chow et al. [2019] projects action or parameters to ensure the decrease of Lyapunov function after a step. Ohnishi et al. [2019] is similar to ours but it constructs a barrier function manually instead of learning such one. Ames et al. [2019] gives an excellent overview of control barrier functions and how to design them. Perhaps the most related work to ours is Cheng et al. [2019], which also uses a barrier function to safeguard exploration and uses a reinforcement learning algorithm to learn a policy. However, the key difference is that we *learn* a barrier function, while Cheng et al. [2019] handcrafts one. The works on Lyapunov functions [Berkenkamp et al., 2017, Richards et al., 2018] require the discretizating the state space and thus only work for low-dimensional space.

# 8 Conclusion

In this paper, we propose a novel algorithm CRABS for training-time safe RL. The key idea is that we co-train a barrier certificate together with the policy to certify viable states, and only explore in the learned viable subset. The empirical rseults show that CRABS can learn some tasks without a single safety violation. We consider using model-based policy optimization techniques to improve the total rewards and sample efficiency as a promising future work. Another fascinating direction is how to deal with less accurate learned dynamics model in higher dimension environments.

## Acknowledgement

We thank Changliu Liu, Kefan Dong, and Garrett Thomas for their insightful comments. YL is supported by NSF, ONR, Simons Foundation, Schmidt Foundation, DARPA and SRC. TM acknowledges support of Google Faculty Award, NSF IIS 2045685, and JD.com.

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
