# OpenReview forum: "Learning Barrier Certificates: Towards Safe Reinforcement Learning with Zero Training-time Violations"
_NeurIPS.cc/2021/Conference — NeurIPS 2021 Poster_

### Official Review · Reviewer_78sq · 2021-07-13

**Rating:** 6
**Confidence:** 3

**Summary:**

The paper proposes an RL algorithm that starts from an initial safe but low return policy that is iteratively improved while ensuring no safety violation occurs during the rollouts required by the training. The paper achieves this by learning barrier functions as safety certificates.  The two "simpler" barrier conditions are directly enforced in the structure of the learned function. In contrast, the "closed under a policy+environment step" condition R3 is checked heuristically on a probabilistic model of the environment with a Metropolis-adjusted Langevin algorithm.
Finally, the authors evaluate the achieved return and number of safety violations of their RL algorithm and other RL algorithms on variations of the Pendulum and CartPole gym environments.

**Limitations And Societal Impact:**

I think most of the limitations of the paper are sufficiently communicated (except my first concern in the main review).

**Main Review:**

I think the paper is well written.
Nonetheless, I noticed a few unclear parts of the paper that I would like to authors to clarify.

To check R3 requires solving the non-convex/concave global optimization problem in Eq. (1) and its derived forms. As MALA works only with a set of local maxima, no guarantees about the global optimality can be made (it seems to be only a heuristic). I think this design choice is okay but needs to be communicated more clearly, i.e., as it is very well done in section 4.3 that there is a tradeoff between feasibility and correctness of the transition model (in the sense that a flawed dynamics model can lead to incorrect safety guarantees of the barrier function).


The section about the handcrafted barrier function is a bit imprecise. As a barrier function is policy dependent, I suppose the "handcrafted" function is a barrier function of the initial safe policy. Therefore, I would then expect that the achievable return when training the initial policy very much depends on the specific barrier certificate, i.e., if by the initial policy non-reachable but actually by a better policy reachable safe states are considered safe or unsafe.
Moreover, how does the scenario with a handcrafted barrier function lead to safety violations? This shouldn't be the case? Please elaborate.

In Figure 2 top row, it seems that CRABS learns faster than unconstraint SAC. This is counter-intuitive and suggests that the hyperparameters are inadequately chosen, i.e., either SAC hyperparameters are non-optimal or CRABS hyperparameters received a much larger tuning budget.

I like that the authors discuss the limitations of their approach with respect to higher dimensional problems, i.e., that learning a dynamics model becomes challenging and leads to safety violations.

I positively acknowledge that the authors provide the code for their experiments.


I am looking forward to reading the author's response.

**Time Spent Reviewing:**

4

---

> ### Author Response · Authors · 2021-08-11
> **Review Response**
>
> We thank the reviewer for the comments and for noting “the paper is well written”. We’ll address the concerns below and fix minor issues in the revision.
>
> ## Clarification on global optimality in MALA
> Thanks for the suggestion. We’ll make it clearer that MALA may not empirically find a global optimum in the revision. We will also clarify in the revision that an optimization problem can be viewed as a sampling problem, as sampling from $\exp(\tau f(x))$ (with $\tau$ a hyperparameter) yields an approximation of the global maximum of a function $f$. Theoretical works [1] have shown that under mild regularity assumptions, the stationary distribution of MALA is the distribution that we want to sample from.
>
> ## More details on handcrafted barrier certificates
> We’d like to make a clarification on our experiments here. Our goal is to only show that it’s non-trivial to find a subset of viable states by handcrafting a region and therefore to demonstrate the necessity to learn the barrier certificates. In the experiments, we handcrafted a function $g$, whose superlevel set is a subset of safe states, and restricted exploration in this region in a similar manner as our current exploration scheme (Algorithm 3). Here $g$ only satisfies R1 and R2 but not necessarily R3, and the design of $g$ doesn’t depend on the initial policy (But indeed the full trajectory of the initial policy is contained in the region.) In other words, strictly speaking, the designed function g is not a barrier certificate, and it only merely attempts to characterize a subset of safe states and restrict the exploration to it. But mostly because there is no guarantee that the superlevel set of $g$ excludes all irrecoverable states, as shown in the experiment, using $g$ as the poor man’s version of the barrier certificate doesn’t work at all and still leads to safety violations. We will clarify our terminology and experiments in the revision.
>
> Finally, we note that designing a barrier certificate that satisfies all of R1,R2, and R3 is a challenging task that has been extensively studied in the control theory literature [2]. However, existing works oftentimes require prior knowledge about the dynamics. Our contribution is to *learn* the barrier certificates with adversarial training, without leveraging prior knowledge about the dynamics.
>
> ## Why can CRABS learn faster than unconstrained SAC?
> Regarding hyperparameter tuning: For CRABS, we didn’t tune the policy improvement algorithm (which is SAC) but simply use the default hyperparameters. In fact, unconstrained SAC requires one more hyperparameter: the penalty when it visits unsafe states. As stated in Appendix C.3, we tune this hyperparameter in [3, 10, 30, 100] and choose the one that works best for unconstrained SAC. We found this hyperparameter unnecessary in CRABS because it never visits unsafe states.
>
> Indeed, it may appear surprising that CRABS learns faster than unconstrained SAC. Our explanation will be that the training-time safety in CRABS makes it easier to optimize the policy, and we’ll give two perspectives of why this is the case.
> 1. The value function or Q-function in CRABS is more smooth. We discuss the value function for simplicity. Suppose we have two nearby states $s_1$ and $s_2$, where $s_1$ is safe and $s_2$ is unsafe.  Under SAC, the values $V^*(s_1)$ and $V^*(s_2)$ need to differ by a lot because where $s_1$ is safe and $s_2$ is unsafe, and therefore $V^*$ is non-smooth and SAC might be slow to learn it. In contrast, CRABS doesn’t care about the value of $V^*(s_2)$ because CRABS only sees safe states (and never sees $s_2$). Therefore the policy and value optimization in CRABS could be easier.
> 2. Zero training-time violation provides samples with higher quality. The collected samples in CRABS only need to focus on the reward, as safety is guaranteed by the exploration scheme. That is, given the same number of samples, some of the collected samples in unconstrained SAC are spent on learning to be safe, while all collected samples in CRABS can be devoted to improving the total rewards.
>
> [1]: Robert, Christian, and George Casella. Monte Carlo statistical methods. Springer Science & Business Media, 2013.
>
> [2]:  ​​Ames, Aaron D., et al. "Control barrier functions: Theory and applications." 2019 18th European Control Conference (ECC). IEEE, 2019.

---

### Official Review · Reviewer_DXuC · 2021-07-16

**Rating:** 7
**Confidence:** 4

**Summary:**

This work proposes an algorithm of iteratively learning dynamical system model, barrier certificate and policies, for safe RL.

**Ethical Concerns:**

The reviewer believes there is no significant ethical concern in this work.

**Limitations And Societal Impact:**

Limitations are well discussed.  However, as long as the author(s) stress zero training-time violaton, the limitation is crucial.

**Main Review:**

The overall idea of co-learning of dynamics, barrier certificate and policy are interesting and important.
And the paper is clearly written.
However the reviewer has the following concerns.

Major concerns:
1) The author(s) mention that the proposed method potentially works for higher dimensional systems; however the experiment is only considering rather simple models.
Also, the work stresses that the proposed method aims at zero training-time violations and compared it against existing safe RL work; however, as the author(s) also mention in the experiment section that more data are necessary to achieve zero violations, safety violations ultimately depend on data, representational capacity, disturbance, convergence of algorithm, approximation of confidence interval etc etc. in practice.  Note theoretically it is easy to say zero-training violation if one makes certain assumptions.  Therefore, it is very important to clearly discuss why the proposed method outperforms others in terms of safety violations from several perspectives.

2) There is already an existing work that does co-learning of dynamics, Lyapunov function (through value function learning), (and policy). (e.g. Berkenkamp)  This work basically replaced Lyapunov function by barrier certificate, and generalized the problem setting.
(please consider the nominal known dynamical system model as that can be obtained by initial safe policies assumed in this work)
Therefore, it is again very important to discuss why the use of barrier certificate with the proposed adversarial training may be better than that.

If the above concerns are addressed, the reviewer is happy to raise the score.

----------
review updated after the initial response

**Time Spent Reviewing:**

6 hours

---

> ### Author Response · Authors · 2021-08-11
> **Review Response**
>
> We thank the reviewer for the comments and for finding the overall idea “interesting and important”. We will address the major concerns below.
>
> ## More challenging environment
> > The author(s) mention that the proposed method potentially works for higher dimensional systems; however the experiment is only considering rather simple models.
>
> We would like to clarify that these environments, though only have 2-4 dimensional state space, are already very challenging for zero training-time violation because they are designed such that the high-reward regions are also high-risk (closer to the safety boundary). The exploration scheme must balance well between safety and high rewards.  All baseline algorithms require > 100 violations. We would like to respectfully argue that given the fact that the baselines have many violations, the importance of the safe RL, and the existence of many real-world low-dimensional RL problems, we believe that it’s still valuable to make progress for low-dimensional state space problems with zero violations. Arguably, for such a challenging and critical question, the research community may likely need to start with simple environments and gradually make progress.
>
>
> ## Discuss if CRABS outperforms others in terms of safety violations from several perspectives
> We compare CRABS with other methods from several perspectives. If there are any particular perspectives that the reviewer might be interested in, we respectfully ask the reviewer to let us know.
> 1. Safety violations. Our main metric is the number of safety violations and we outperform the baselines, as shown in the experiments. Our theory shows zero violations if we assume each module works perfectly (which might be not realistic as the reviewer kindly pointed out), but the RL algorithms that we compare with generally do not provide zero violations guarantees under any assumptions. As we discussed in the introduction, many existing RL works learn by contrasting safe and unsafe trajectories, so they can’t achieve zero training-time safety. (Other algorithms with zero violations guarantees [1,3] along with the control theory literature oftentimes assume more prior knowledge about the dynamics)
> 2. Total sample efficiency. Indeed, our algorithms might require more samples than other algorithms, and this is mostly due to the fact that all our samples are safe trajectories, which lack diversity. We believe fundamentally we need to collect more samples, if the samples are only allowed to be safe samples.
> 3. Modularity: CRABS can be easier to debug, as it decomposes the algorithm into many verifiable components. Even if CRABS fails, we can attribute the failure to the inaccurate dynamics model so we should improve the dynamics model, or to the incorrect barrier certificates, whose correctness can be verified by additional computational power without any samples from the environment. It’s more difficult to do so for methods based on extrapolation of Q-functions (e.g., RecoveryRL and SQRL compared in our experiments), as understanding the extrapolation of learned Q-function is a difficult problem in RL.
>
> ## Why is the proposed method better than Berkenkamp et al. [1] and Richards et al. [2]?
> We thank the reviewer for pointing out this very important related work. Richards et al. is a follow-up work of Berkenkamp et al. and therefore here we mostly focus on the comparison with Richards et al. We also refer to the reviewer to Line 53-64 in the introduction for a more concise comparison. We will also incorporate the discussion below in the revision.
> 1. CRABS does not need to discretize the state space whereas Berkenkamp et al. do.  Therefore we can work with state dimension = 4 and might be able to extend to higher dimensional state space, whereas Berkenkamp et al. and Richards et al. have only experiments in 2-dimensional state space. In fact, Berkenkamp et al. discretize the state space and make use of the discretized state space heavily (compute the value function, find a Lyapunov function, policy optimization, etc.). We consider removing the need for state discretization important and non-trivial.
> 2. Barrier certificates are generally less restrictive than Lyapunov functions, as Lyapunov functions must be decreased at each step, so they are incapable of certifying the safety of some policies, e.g., those which make circular movements. (But we didn’t propose barrier certificates. What’s new in our work is the adversarial training algorithms for learning the barrier certificates. In contrast, most prior works design barrier certificates.)
> 3. As mentioned by the reviewer, Berkenkamp et al. use the (discounted) cost-to-go (i.e., the value function) to be the Lyapunov function. This seems to be a reasonable choice, but there’s no guarantee that it’s a valid Lyapunov function. In theory, only *undiscounted* value functions are guaranteed to be Lyapunov functions. Furthermore, Berkenkamp et al. assume a positive reward function. Learning an accurate value function itself is also a very challenging task.
> 4. Richards et al. use the ground-truth dynamics model, while our algorithm learns iteratively the dynamics model and grows it during training. The use of an imperfect learned dynamics model is also an important topic for many RL algorithms, while Richards et al. don’t address it.
> 5. Richards et al. only certify the safety of some states without learning a policy, while our algorithm also learns the policy and tailors the certificates to the learned policies. The challenge of balancing the exploration and exploitation, which is fundamental in RL, is not addressed by Richards et al.
>
>
> [1]: Berkenkamp, Felix, et al. "Safe model-based reinforcement learning with stability guarantees." Proceedings of the 31st International Conference on Neural Information Processing Systems. 2017.
>
> [2]: Richards, Spencer M., Felix Berkenkamp, and Andreas Krause. "The lyapunov neural network: Adaptive stability certification for safe learning of dynamical systems." Conference on Robot Learning. PMLR, 2018.
>
> [3]: Cheng, Richard, et al. "End-to-end safe reinforcement learning through barrier functions for safety-critical continuous control tasks." Proceedings of the AAAI Conference on Artificial Intelligence. Vol. 33. No. 01. 2019.

---

> > ### Comment · Reviewer_DXuC · 2021-08-19
> > **After the initial response**
> >
> > Thank you for the response; the response is perfectly addressing my concerns.  The original manuscript was a bit unclear about how the proposed method is better at zero training error compared to other methods.  It should be helpful to add some table showing what assumptions are needed to guarantee zero-training error for existing work and the proposed work.  (such as discrete state, realizability of the function class, prior safe trajectory, prior nominal model, undiscounted value function can be obtained, barrier certificate can be obtained, sample size etc.etc.)
> >
> > Assuming the author(s) will provide these clarifying statements in the final version, the reviewer raises the score to 7 from 5.

---

> > > ### Author Response · Authors · 2021-08-20
> > > **Thanks for the consideration of our response**
> > >
> > > We thank the reviewer for the consideration of our response and the suggestions to include the comparisons.  We will clarify the assumptions of our methods and prior works in the next revision as requested by the reviewer.
> > >
> > > For the reviewer’s convenience, we list here the key assumptions for a few prior methods and ours. We will incorporate these into the final version.
> > >
> > > Berkenkamp et al. + Richards et al.  assume
> > > 1. Discretization of state space (partially removable in Richards et al.)
> > > 2. Calibrated dynamics model (learned from data).
> > > 3. An initial safe policy.
> > > 4. A positive reward function.
> > > 5. An accurate (discounted) value function.
> > > 6. Solving a constrained optimization problem (via Lagrangian multipliers).
> > >
> > > Srinivasan et al. assume
> > > 1. No state is irrecoverable.
> > > 2. A diverse and large dataset for pretraining
> > > 3. The pretrained Q-function is optimal (extrapolating to unseen state-action pairs).
> > >
> > > CRABS assumes
> > > 1. Calibrated dynamics model (learned from data).
> > > 2. Extrapolation of the dynamics to nearby unseen states.
> > > 3. An initial safe policy.
> > > 4. Solving a minimax optimization problem with adversarial training.
> > > 5. Realizability of function class (function class contains barrier function).

---

### Official Review · Reviewer_SUah · 2021-07-16

**Rating:** 6
**Confidence:** 4

**Summary:**

This paper proposes a safe reinforcement learning algorithm, CRABS, which can achieve zero training-time safety violations. The key idea is to iteratively learn the dynamic model, the policy and the barrier certificates. The barrier certificates are learned via adversarial training and can constrain the exploration of the policy in the safe set. The proposed algorithm is evaluated in four simple classical control problems (variants of inverted pendulum and cartpole). The evaluations demonstrate zero training-time violations and comparable learning performance compared to other safe learning baselines.

**Limitations And Societal Impact:**

Both the limitation and the potential societal impact are discussed.

A suggestion: I like Appendix E, which is a comprehensive summary of the limitations. It would be great to make limitation more explicit in the main text, instead of "hidden" in the Appendix.

**Main Review:**

Safe learning with zero training-time violations is critical for applying reinforcement learning to real-world systems, such as robotics. This paper proposes a method that nicely combines reinforcement learning (SAC) with control theory (barrier certificates) to tackle this exact challenge. The paper is well written. The method is novel. The results are promising.

Although I like the proposed method and I believe that it has good technical contributions, I did not give the paper a high score mainly due to the evaluations. I am disappointed that the method is only evaluated on simple control problems based on inverted pendulum and cartpole. Although it seems that CRABS can beat baselines (CPO, recovery RL) in these environments, it is not evaluated in more complicated robotic environments, such as locomotion, navigation, and manipulation, where the baselines work well. To make CRABS useful and impactful, experiments that show CRABS can work equally well or better in complex environments are essential.

Here are a few detailed suggestions how the paper can be improved:
1) It is important to include more evaluations on challenging environments and show the advantages of CRABS. I understand that the model-learning part is the bottleneck to scale up the method to high-dimensional environments. Since for many control problems, the model is well known (e.g. the unicycle model for wheeled robots), it is still valuable to show that if the ground-truth or an approximate model is given, CRABS can guarantee safety during the training process. A navigation task with wheeled robots (safety gym) seems to be a good example to add.

2) It appears to me that CRABS learns a model and uses that model in barrier certificate, but does not leverage the model in policy learning. The policy learning is a modified SAC, a model-free method. As we all know, model-based RL is more data efficient than its model-free counterparts. I wonder whether the policy update of CRABS can use the learned model, and then becomes more data efficient. This is just a suggestion for the long-term research direction, not a criticism for the current paper.

3) The second paragraph of 2.2 seems incorrect to me. According to the text, If h(s)>=0 and h(s')>=0, and then h is a barrier certificate. If this is the case, finding a barrier certificate would be easy: h = |s|^2. Did I miss anything?

4) Throughout of the paper, there are a few equations that have argmax_{s:h_\phi(s)<=1} U. I am not sure where "h_\phi(s)<=1" comes from. Shouldn't it be h_\phi(s)<=0?

5) Since the effectiveness of CRABS heavily depends on the model accuracy, it would be nice to have an ablation study to understand the relation between the number of training-time violations and the model accuracy.

6) It would be great to give more details about "Handcrafted barrier function" paragraph in Section 6. Since how the barrier certificate is handcrafted would significantly impact the performance of the algorithm, the conclusion that "this variant does not perform well" is less convincing without details of handcrafting.

7) Typo: Line 352: rseults->results

**Time Spent Reviewing:**

3 hours

---

> ### Author Response · Authors · 2021-08-11
> **Review Response**
>
> We thank the reviewer for the comments and for noting our “good technical contribution”. We will address the major concerns below and fix the minor issues in the revision.
>
> ## More evaluations on challenging environments
> Indeed CRABS is only evaluated in relatively simple environments. We would like to clarify that these environments, though only have 2-4 dimensional state space, are already very challenging for zero training-time violation because they are designed such that the high-reward regions are also high-risk (closer to the safety boundary). The exploration scheme must balance well between safety and high rewards.  All baseline algorithms require > 100 violations. We would like to respectfully argue that given the fact that the baselines have many violations, the importance of the safe RL, and the existence of many real-world low-dimensional RL problems, we believe that it’s still valuable to make progress for low-dimensional state space problems with zero violations. Arguably, for such a challenging and critical question, the research community may likely need to start with simple environments and gradually make progress.
>
> We thank the reviewer for the suggestions of running on higher dimensional state space with accurate dynamics. We are in the process of finding suitable environments (where the dynamics are known and differentiable) and testing our algorithms. We will update our results as soon as they are available.
>
> ## Model-based policy optimization
> We thank the reviewer for the great suggestion that model-based policy optimization can be a future direction. We have a similar idea and have already included it as promising future work in Section 8. We decided to use a model-free planner in our current algorithm mostly because we would like to be able to make sure the source of the improvement only stems from the use of barrier certificates.
>
> ## The theory in paragraph 2.2
> > The second paragraph of 2.2 seems incorrect to me. According to the text, If h(s)>=0 and h(s')>=0, and then h is a barrier certificate. If this is the case, finding a barrier certificate would be easy: h = |s|^2.
>
> The most general version of the barrier certificate, as stated in the paragraph, is used to certify that the trajectory won’t leave a set once it enters the set. The set is the superlevel set of the barrier certificate. Your example is indeed a barrier certificate, but the certified superlevel set $\\{s: h(s) \geq 0\\}$ is the whole state space, so it is an uninteresting barrier certificate. To make the barrier certificate more meaningful, additional requirements are needed. E.g., to ensure safety, at the beginning of Section 3, we need the barrier certificate to satisfy R1 and R2 in addition to R3.
>
> ## Ablation study
> > it would be nice to have an ablation study to understand the relation between the number of training-time violations and the model accuracy.
>
> Thanks for the suggestions. However, It’s difficult to ablate the relationship between the number of training-time violations and model accuracy because our algorithm is designed only for zero violations, not for a tradeoff between the model accuracy and the number of safety violations. If the model is not accurate enough, we can’t find a good barrier certificate to certify a policy, and the main algorithm (which trains the dynamics model, policy, barrier certificates iteratively) can’t proceed.
>
> ## Handcrafted barrier certificates
> We’d like to make a clarification on our experiments here. Our goal is to only show that it’s non-trivial to find a subset of viable states by handcrafting a region and therefore to demonstrate the necessity to learn the barrier certificates. In the experiments, we handcrafted a function $g$, whose superlevel set is a subset of safe states, and restricted exploration in this region (the superlevel set of $g$) in a similar manner as our current exploration scheme (Algorithm 3). Here $g$ only satisfies R1 and R2 but not necessarily R3.  Therefore, strictly speaking, the designed function g is not a barrier certificate, but only merely attempts to characterize a subset of safe states and restrict the exploration to it. (But indeed the full trajectory of the initial policy is contained in the region.) But mostly because there is no guarantee that the superlevel set of $g$ excludes all irrecoverable states, as shown in the experiment, using $g$ as the poor man’s version of the barrier certificate doesn’t work at all. We will clarify our terminology and experiments in the revision.
>
> More low-level detail: the handcrafted function $g$ takes the form of $g(s) = 1 - \max(\omega(x / x_\text{max}), \omega(\theta / \theta_\text{max}), \omega(\dot x / \dot x_\text{max}), \omega(\dot \theta / \dot \theta_\text{max}))$ with $\omega(x) = \max(0, 100 (|x| - 1))$, and four hyperparameters $x_\text{max},  \dot x_\text{max},  \theta_\text{max},  \dot \theta_\text{max}$. We tried $x_\text{max},  \dot x_\text{max},  \theta_\text{max},  \dot \theta_\text{max}$ to be $(0.5\lambda, 2 \lambda, 0.7 \lambda, 2 \lambda)$ with $\lambda$ chosen from $\\{0.3, 0.7, 1\\}$ and found out that all of them have a similar number of safety violations as unconstrained SAC, or even more.
>
> Finally, we note that designing a barrier certificate that satisfies all of R1,R2, and R3 is a challenging task that has been extensively studied in the control theory literature [1]. However, existing works oftentimes require prior knowledge about the dynamics. Our contribution is to *learn* the barrier certificates with adversarial training, without leveraging prior knowledge about the dynamics.
>
>
> ## Typos
> Thanks for pointing out the typos and sorry for the confusion caused by them. All $h_\phi(s) \leq 1$ should be $h_\phi(s) \leq 0$. We’ll fix them in the revision.
>
>
> [1]: ​​Ames, Aaron D., et al. "Control barrier functions: Theory and applications." 2019 18th European Control Conference (ECC). IEEE, 2019.

---

> > ### Comment · Reviewer_SUah · 2021-08-30
> > **Thanks for the response**
> >
> > The response clarifies most of my questions. Thank you. I still keep the score unchanged due to limited evaluations using simple environments.

---

### Author Response · Authors · 2021-08-11
**To All Reviewers**

We thank all reviewers for their valuable and helpful comments. We’ve addressed the concerns below. We are more than happy to answer if the reviewers have further questions.

---

### Decision · Program_Chairs · 2021-09-27

**Decision:**

Accept (Poster)

**Comment:**

I agree with the reviewers that this paper deserves to be accepted; it has a novel foundational contribution and is well written.

The paper presents a new approach that iteratively generates a barrier certificate, an environment model and a policy, in a way that ensures that there are no safety violations even during training. The algorithm requires an initial safe (but possibly suboptimal) policy, and optimizes it while ensuring that there will not be any safety violations during training.

As pointed out by one of the reviewers, the main downside of the paper is that the experiments are not very ambitious; the approach is evaluated on only a few problems with a very low-dimensional state space. However, the task of training with no safety violations is already challenging even for these relatively simple environments. There are also a few places where the explanations could be a little clearer, for example, the way algorithm 2 is presented, it is not entirely clear where the initial barrier certificate comes from. But overall, this is a very strong paper with a solid contribution to the state of the art in safe reinforcement learning.